… PLOS ONE

# Evaluation of the effectiveness of macaíba palm seed kernel (*Acrocomia intumescens* drude) on anxiolytic activity, memory preservation and oxidative stress in the brain of dyslipidemic rats

**Roberta Cristina de França Silva**[1]\*, **Mikaelle Albuquerque de Souza**[2], **Jaielison Yandro Pereira da Silva** [1], **Carolina da Silva Ponciano**[1], **Vanessa Bordin Viera**[1], **Camila Carolina de Menezes Santos Bertozzo**[1], **Gerlane Coelho Guerra**[3], **Daline Fernandes de Souza Araújo**[3], **Marta Maria da Conceição**[2], **Celina de Castro Querino Dias**[2], **Maria Elieidy Oliveira**[2], **Juliana Kessia Barbosa Soares** [1,2]

**1** Universidade Federal de Campina Grande, Cuité, PB, Brasil, **2** Universidade Federal da Paraíba, João Pessoa, PB, Brasil, **3** Universidade Federal do Rio Grande do Norte, Natal, RN, Brasil

\* robertasaron@gmail.com

## Abstract

Macaíba palm seed kernel is a source of lipids and phenolic compounds. The objective of this study was to evaluate the effects of macaíba palm seed kernel on anxiety, memory, and oxidative stress in the brain of health and dyslipidemic rats. Forty rats were used, divided into 4 groups (n = 10 each): control (CONT), dyslipidemic (DG), kernel (KG), and Dyslipidemic kernel (DKG). Dyslipidemia was induced using a high fat emulsion for 14 days before treatment. KG and DKG received 1000 mg/kg of macaíba palm seed kernel per gavage for 28 days. After treatment, anxiety tests were carried out using the Open Field Test (OFT), Elevated Plus Maze (EPM), and the Object Recognition Test (ORT) to assess memory. In the animals' brain tissue, levels of malondialdehyde (MDA) and total glutathione (GSH) were quantified to determine oxidative stress. The data were treated with Two Way ANOVA followed by Tukey (p <0.05). Results demonstrated that the animals treated with kernel realized more rearing. DG and KG groomed less compared with CONT and DKG compared with all groups in OFT. KG spent more time in aversive open arms compared with CONT and DKG compared with all groups in EPM. Only DKG spent more time in the central area in EMP. KG and DKG showed a reduction in the exploration rate and MDA values (p <0.05). Data showed that macaíba palm seed kernel consumption induced anxiolytic-like behaviour and decreased lipids peroxidation in rats' brains. On the other hand, this consumption by healthy and dyslipidemic animals compromises memory.

**Data Availability Statement:** All relevant data are within the paper and supporting information files.

**Funding:** This study was supported by CNPQ (National Council for Scientific and Technological Development) Brazil - Universal 423166/2018-2. The funders had no role in study design, data collection and analysis, decision to publish, or preparation of the manuscript. JKBS is the researcher responsible for using the funding. There was no additional external funding received for this study.

**Competing interests:** The authors have declared that no competing interests exist.

## Introduction

Lipids are increasingly recognized for their role in brain function, especially membrane lipids involved in cell signaling. As part of neuronal composition, lipids can influence subjective perception, mood, and emotional behavior. Long-term changes in diet can influence lipid composition in the brain, and can directly affect the release of neurotransmitters and neurotrophins from presynaptic membranes, influencing both cognitive and motor functions. The anxiolytic/antidepressant effects of diet may result directly from an increase in the supply of lipids which are involved in synaptic plasticity, preserving and improving cognitive functions [1–4].

A study involving different types of unsaturated lipids yielded protective effects for neural tissues, preventing behavioral changes including depression and anxiety disorders in the treated animals [5]. In animals treated with olive oil, was observed decreases in anxiety and depressive behavior, in addition to long-term memory improvements [6]. A similar study treating animals with fish oil reported improvements in anxiety, depression, and cognition behaviors [7].

Yet when consumed in excess, lipids can lead to metabolic disorders such as obesity which is associated with brain disorders, and cognitive decline, a manifestation of neurodegenerative disease [8–10]. An experimental study involving administration of palmitic acid, a saturated fatty acid, augmented anxiety-like behavior in animals [11]. Animals fed high-fat diets (HFD) reveal reduced levels of brain-derived neurotrophic factor in the hippocampus, reduced neurogenesis, and impaired learning as compared to animals fed standard diets [12–14]. Changes in hypothalamic-pituitary-adrenal axis functions that comprise cognitive and behavioral functions are often due to excessive consumption of lipids, causing deficiencies in emotional processes and in stress response [15]. However, we do not know if dyslipidemia can induce brain changes and how the consumption of food source of lipids and antioxidants compounds can reverse possible damage.

Vegetable lipids present bioactive compounds and numerous associated properties of these natural compounds in seed oils, we note phenolic compounds and carotenoids [16,17]. An increasing number of studies associate the potential of lipids derived from seeds and kernels with protective effects on the neurological system. Kernel, found in tropical regions, are rich in plant diversity, and especially favor the discovery of new active ingredients [18–20].

Macaíba is the fruit of the macaibeira (*Acrocomia intumescens* Drude), a palm tree that naturally occurs in northeastern Brazil. The fruit is typical of the region with exotic and intense flavors and aromas. It presents a hard surrounding outer shell protecting a secondary starch layer (mesocarp or pulp), and at the center of the fruit, a hard endocarp occurs with one or two kernels. Kernel contains about 27% of lipid and a higher concentration of total phenolics [21–23].

There are few studies assessing the potential and quality of macaíba pulp and oil [22,24]. A research confirmed the anti-inflammatory and diuretic activities of the pulp's oil in rats [25]. However, there is no scientific evidence of the effect in human or animal health induced by the consumption of macaíba palm seed kernel.

Based on the above, it is expected that the macaíba palm seed kernel will present protective and antioxidant effects, with anxiolytic activity and memory preservation in health rats or with dyslipidemia. The aim of this study was to evaluate the effects of macaíba palm seed kernel on anxiety behavior and short-term memory in adult health and dyslipidemic Wistar rats.

## Materials and methods

### Seed kernel extraction from macaíba palm

The macaibeira fruits were obtained from palm trees located in the city of Areia, -6.963845 / -35.749738, state of Paraíba, northeast region of Brazil. Registration in the national system for

the management of genetic heritage and associated traditional knowledge (SISGEN): ADD854A.

To obtain macaíba palm seed kernel flour, the peel and pulp was removed from the macaíba fruit, chestnut was broken and kernel were removed. After obtaining the kernel, the films surrounding the kernel were removed and taken to grind to obtain the flour. The flour was taken to a drying oven with air circulation at 55˚C (± 1˚ C) during 24 hour. The flour was sieved to eliminate larger grains; then stored in a vacuum and kept on refrigeration until the moment of use.

## Determination (macaíba palm seed kernel) phenolic compounds and total antioxidant activity

**Extraction.** The constituents of the macaíba palm seed kernel were extracted with 80% of ethanol and evaluated for ABTS removal capacity, iron reducing activity (FRAP), total phenolic and total flavonoids. One gram of macaíba palm seed kernel was inserted into a test tube and 10 ml of solvent was added. The test tube was left at room temperature for 60 minutes, and after filtration the volume was completed to 10 ml with the extraction solvent and stored in a freezer (-18˚C) until analysis. All extractions were performed in triplicate.

**Determination of total phenolic compounds (TPC).** To estimate the total phenolic compounds, the methodology described by Liu et al. [26] was used with minor modifications. Briefly, 250 μl of extract was mixed with 1250 μl of a 1:10 diluted Folin–Ciocalteu reagent. The solutions were mixed well and incubated at room temperature (27˚C) for 6 minutes. Then, 1000 μl of 7.5% sodium carbonate solution ($Na_2CO_3$) was added, and again incubated in a bath at 50˚C for 5 minutes. The absorbance of the reaction mixtures was measured at 765 nm using a spectrophotometer (BEL Photonics, Piracicaba, São Paulo, Brazil). The absorbance of the extract was compared with a standard curve of gallic acid to estimate the total phenolic compound (TPC) concentration in the sample. TPC were expressed in mg of gallic acid equivalents (GAE) per hundred grams of macaíba palm seed kernel based on dry weight (DW).

**Determination of total flavonoids.** The total flavonoid content was measured using the colorimetric assay developed by Zhishen et al. [27]. A known volume (0.5 ml) of the extract was added to a test tube and 150 μl of 5% $NaNO_2$ was added. After 5 minutes, 150 μL of 10% $AlCl_3$ was added; and after 6 minutes, 1 mL of NaOH 1 M, followed by the addition of 1.2 mL of distilled water. The sample absorbance was read at 510 nm spectrophotometry (BEL Photonics, Piracicaba, São Paulo, Brazil). The absorbance of the extract was compared with a standard catechin curve to estimate the concentration of flavonoid contents in the sample. The flavonoid content was expressed in mg of catechin equivalents (QE) per hundred grams of macaíba palm seed kernel based on dry weight (DW).

**Determination of total carotenoids.** Total carotenoids were determined by the Higby method [28]. The extracts were prepared using 1 g of macaíba palm seed kernel macerated in 10 mL of hexane PA and calcium carbonate, remaining protected from light for 12 hours under refrigeration. Subsequently, centrifugation was performed at 5724 xg for 10 minutes for later reading using a spectrophotometer (BEL Photonics, Piracicaba, São Paulo, Brazil) at 450 nm. The kernel extracts were previously macerated with 50% acetone, which was discarded, and calcined sand was added. The results were calculated using the formula: Total carotenoids = (A450 x 100)/(250 x L x W), where A450 = absorbance; L = width of the cuvette in cm; and W = quotient of the sample mass in grams to the final dilution volume in mL.

**Determination of total yellow flavonoids.** Total yellow flavonoids were determined according to the method of Francis [29]. The extracts were prepared using 1 g of the macaíba palm seed kernel in 10 ml of extractive 95% ethanol/1.5 NHCl (85:15) solution, remaining protected from light for 12 hours under refrigeration. Afterwards, centrifugation was performed

at 7244 xg for 15 minutes, for later reading with a spectrophotometer (BEL Photonics, Piraci-caba, São Paulo, Brazil) at 374 nm. The results were calculated using the formula: dilution fac-tor x absorbance/76.6.

**Antioxidant activity-FRAP method.** The FRAP method was carried out according to Benzie and Strain [30], with modifications proposed by Pulido et al. [31]. In this assay, 3.6 mL of FRAP reagent (0.3 M, pH 3.6 acetate buffer, 10 mM TPTZ (2,4,6-Tris(2-pyridyl)-s-triazine), and 20 mM ferric chloride) were mixed with 200 μl of diluted extract in distilled water and incubated for 30 minutes at 37˚C. The FRAP solution was used as a reference reagent, and the absorbance was read with a spectrophotometer (BEL Photonics, Piracicaba, São Paulo, Brazil) at 593 nm. The results were expressed in μmol of trolox equivalents per gram of macaíba palm seed kernel on a dry weight basis (DW) (μmol TE/g-1).

**Antioxidant activity-ABTS method.** The ABTS (2,2'-azino-bis (3-ethylbenzothiazoline-6-sulfonic acid) diammonium salt) method was performed according to the methodology described by Surveswaran et al. [32], with modifications. The ABTS radical was formed from a reaction of 140 mM potassium persulfate with a 7 mM ABTS (2,2′-azino-bis(3-ethylbenzothia-zoline-6-sulfonic acid) diammonium salt stock solution, kept in the dark at room temperature for 16 h. For analysis, the ABTS radical was diluted in distilled water until a solution with an absorbance of 700nm ± 0.02 nm at 734 nm was obtained. A 100μL aliquot of each extract was homogenized with 500μL of the ABTS radical. The absorbance of the samples was reading with a spectrophotometer (BEL Photonics, Piracicaba, São Paulo, Brazil) at 734 nm after 6 minutes of reaction. The results were expressed in μmol of trolox equivalent per gram of macaíba in dry weight (DW) (μmol TE/g$^{-1}$). Where $A_0$ is the absorbance of the control, and for the absorbance of the sample, the effective concentration revealed a 50% radical inhibition activity ($IC_{50}$), expressed in mg extract/mL, which was determined from the graph of free radi-cal scavenging activity (%) against the concentration of the extract.

## Animals and diet

In this study, forty male Wistar rats weighing 200-250g were used from the breeding facility of the Nutrition department of the Federal University of Pernambuco. The animals were housed in indi-vidual metabolic cages, with controlled room temperature (22 ± 1˚C), a constant light-dark cycle (12 hours each), humidity of ± 65%, and receiving feed and water ad libitum. Four groups were formed (n = 10) according to the treatments: 1) Control Group (CONT)—treated with distilled water; 2) dyslipidemic group (DG)—receiving a high fat emulsion (HFE) and distilled water; 3) kernel group (KG)—receiving distilled water, and an macaíba palm seed kernel; and 4) dyslipi-demic kernel group (DKG)—receiving a HFE, and macaíba palm seed kernel. Dyslipidemic administrations in the DG and DKG dyslipidemic experimental groups began two weeks before starting treatment with macaíba. The CONT and KG groups received distilled water in the same proportion in this period; both administered for 14 days through an esophageal tube. Macaíba palm seed kernel (1000 mg/kg of animal weight) was administered daily in the KG and DKG groups, while CONT and DG received distilled water in the same proportion through an esoph-ageal tube, during 4 weeks. All animals had access to food and water *ad libitum*. The experimental protocol followed the ethical recommendations of the National Health Institute Bethesda (BETHESDA, USA) for care and use of experimental animals, and was approved by the Ethics Committee for Animal Use of the Federal University of Campina Grande #057–2016.

## Induction of dyslipidemia

Before starting treatment, dyslipidemia according to the methodology adapted from Xu et al. [33] was induced in the dyslipidemic groups, by means of an HFE. To prepare a 420 ml

solution (HFE), we used: 168 g of lard, 4.70 g of cholesterol, 32.5 g of pasteurized powdered egg yolk (Salto's ®), 8.4 g of bile acid, 42 ml of glycerol, 4.2 g of propiltiouracil and warm distilled water to complete the volume of 420 ml. The solution was stored in the refrigerator and before use heated in a water bath at 42˚ C daily.

The HFE was administered through gavage (at 10 ml/kg of rat weight), once a day, for 14 days prior to the start of treatment. After this period, the animals no longer received HFE.

## Behavioral tests

**Open field test.** At 48 hours from the end of treatments, all animals were submitted to anxiety testing using the open field device. With a 40-watt incandescent lamp at its center, suspended from a height of 46 cm from the floor, the "open field" consists of a circular metal arena in white, about 1 meter in diameter, surrounded by a wall (40 cm height), and the floor is subdivided into 17 fields; 3 concentric circles (respectively: 15, 34, and 55 cm in diameter), subdivided into 16 segments, and a central circle. The open field is a test used to assess anxiety behavior and exploratory activity in rats.

Each animal was placed at the center of the open field and observed for 10 minutes. During this period, evaluations of ambulation; number of segment crossings by the animal (all four legs), rearing, and grooming parameters were performed. The device was cleaned with 70% alcohol and paper towels before starting the tests, and after each animal change, the arena was cleaned with 10% alcohol and paper towels. Manipulation of the animals was always performed by the same researcher. The sessions were recorded with a video camera installed on the ceiling.

## Elevated Plus Maze Test (EPM)

Using the Elevated Plus Maze (EPM) test, anxiety behavior was also assessed. Each animal was placed in a maze made of wood in the shape of a cross, elevated from the ground and formed by two arms with walls and two open (perpendicular) arms. The frequency of entries and time spent by the animal in each type of arm is analyzed. For this type of test, it is observed that the animal tends to explore both types of arms, yet entering and staying for longer in the closed arms. When the level of anxiety is higher, the percentage of entries and time spent by the animal in open arms is lower [34,35].

This test was performed with all experimental groups at 24 hours after the ORT. The animal was placed in the center of the device, facing one of the closed arms, where free exploration was allowed for 5 minutes. The following behavioral categories were analyzed: number of entries into the open and closed arms, the time spent in each of the arms, and time spent in the central area. The sessions were filmed in a low light environment with a video camera installed on the ceiling.

## Object recognition test

Twenty-four hours after the OFT, the object recognition test was performed, using the open field apparatus (white metal circular arena, diameter 100 cm, height 46 cm), to assess the animals' short-term memory. The test consists of first getting the animals used to the open field in the absence of any object, where the can animals freely explore the arena for 10 minutes. Afterwards, in the training session, the animal is placed in the open field with object A1 (familiar object) and object A2 (unfamiliar) where exploration is allowed for 10 minutes. After this training session, at 180 minutes the short-term memory test is performed, where object A1 continues, and object A2 is replaced by object A3 (a new object), and an exploration time of 3 minutes. The animal's exploration time involves sniffing and touching the object with its front legs and/or snout [36]. We used heavy plastic objects with different colors and shapes.

Before starting the test, the device, and objects are cleaned with 70% alcohol and after each change of animals and objects, both the device and the objects themselves are cleaned with 10% alcohol and paper towels. All sessions are recorded with a video camera attached to the device's ceiling.

The results are expressed as a percentage of the total exploration time (computed in seconds). Object recognition index determination is based on the proportion of new object exploration to familiar object exploration ($t_{new}/(t_{new} + t_{familiar})$) [37].

### Determination of macaíba palm seed kernel and brain fatty acid profiles

**Lipid extraction.** Two grams (2 g) of macaíba and each brain sample were weighed in a 50 ml Becker (wet sample) and then 30 ml of a chloroform: methanol (2: 1) mixture was added. The contents were then transferred to a deep glass container (the side covered with aluminum foil) and stirred for 2 minutes with the help of a grinder. The mixture was filtered through qualitative filter paper in a 100 ml graduated cylinder with a polished mouth. The vessel walls were then washed with 10 ml of the chloroform: methanol solution which was filtered with the previous volume. The total volume of the filtered extract from the graduated cylinder was recorded with the cylinder closed, and to 20% of the final volume of the filtered extract, 1.5% sodium sulfate was added. The mixture was then stirred with the graduated cylinder closed, and given time for the phases to separate. It was observed that the upper phase was approximately 40% and the lower phase was approximately 60% of the total volume. The volume of the lower phase was recorded, and the upper phase was discarded by suction with a graduated pipette. To quantify the lipids, a 5 ml aliquot of extract (lower phase) was separated with a volumetric pipette and transferred to a previously weighed beaker. This was placed in an oven at 105˚C, so that the solvent mixture could evaporate, taking care that the fat was not degraded by the heat. After cooling in a desiccator, the beaker was weighed, and the weight of fat residue was obtained from the difference [38].

**Fatty acid methylation.** An aliquot of the lipid extract was used, calculated for each sample according to the fat content found in the lipid measurement and performed according to the Folch, Less and Stanley method [38], by adding 1 ml of internal standard (C19: 0) and a saponification solution (KOH). The solution was then heated to reflux for 4 minutes. The esterification solution was added immediately afterwards, returning the solution to heating under reflux for a further 3 minutes. Afterwards, the sample was allowed to cool before subsequent washes with ether, hexane, and distilled water, obtaining an extract (with methyl esters and solvents), which was conditioned in a properly identified amber glass until the solvents were completely dried. After drying, a suspension in 1 ml of hexane was prepared and packed in a flask for further chromatographic analysis. The aliquots of the saponification and esterification solutions were determined according to the methodology described by Hartman and Lago [39].

**Gas chromatography analysis.** The analyses were performed in a gas chromatograph (VARIAN 430-GC, California, USA), coupled to a fused silica capillary column (CP WAX 52 CB, VARIAN, California, USA) with dimensions of 60 m × 0.25 mm; and a 0.25mm film thickness was used with helium as the carrier gas (flow 1ml/minute). The initial oven temperature was 100˚C programmed to reach 240˚C, increasing 2.5˚C per minute for 30 minutes, totaling 86 minutes. The injector temperature was maintained at 250˚C and the detector at 260˚C. Esterified extract, in 1.0 μl aliquots, was injected into the Split/Splitless injector. Chromatograms were recorded using the Galaxie Chromatography Data System software. Fatty acids were identified by comparing the retention times of the methyl esters of the samples with Supelco Mix C4-24/C19 standards. The fatty acid results were quantified by normalizing methyl ester areas and are expressed in percentage per area.

## Oxidative stress markers

At the end of the experimental tests, following a six-hour fast, the animals were anesthetized with ketamine hydrochloride (50mg/kg) and xylazine (20mg/kg), and then euthanized by cardiac puncture. The brain was then removed for analysis to determine total glutathione and malondialdehyde.

**Determination of total glutathione (GSH).** The total content of glutathione (GSH) was quantified as described by Anderson [40]. The brain tissue samples were broken up into small pieces and then homogenized (Ultra Stirrer Homogenizer, Model 80) with 5% trichloroacetic acid in a 1:20 (w / v) ratio, in an ice bath. The homogenates were centrifuged at 9000 xg (15 minutes at 4˚C) and the supernatants used to quantify the total glutathione content. In a 96-well plate, the homogenous supernatant in duplicate, PBS-EDTA buffer and dithiobisnitrobenzoic acid (DTNB) were added, promoting the transformation from reduced glutathione (GSH) to oxidized glutathione (GSSG). With the addition nicotinamide adenine dinucleotide phosphate (NADPH) in each well, there was a reduction in GSSG by the action of glutathione reductase, constituting an essential redox cycle to maintain the integrity of the cell protective system. Total glutathione was measured immediately in a Polaris® microplate reader (Celer Biotecnologia S. A.) at 412 nm. The results were expressed as nmol/g of tissue. All reagents were purchased by Sigma-Aldrich® (St Louis, MO, USA).

## Determination of malondialdehyde (MDA) levels

Malondialdehyde (MDA) concentrations were determined in the animals' brain tissue as described by Esterbauer and Cheeseman [41]. The samples were thawed, chopped and homogenized (Ultra Stirrer Homogenizer, Model 80) with Tris HCl buffer (pH = 7.4) at a 1: 5 (m/v) ratio. The homogenate obtained was centrifuged at 2500 g for 10 minutes at 4˚C, when chromogenic reagent (1-methyl-2-phenylindol 10.3mM and acetonitrile 3:1) and hydrochloric acid (HCl—37%) were then added to the supernatant. Then, placed in a water bath, with agitation, at 45˚ C, for 40 minutes and, subsequently, taken to centrifugation at 2500 xg, for 5 minutes, at 4˚C. The MDA content was calculated through interpolation with a standard 1,1,3,3—tetraethoxypropane (10mM) curve, hydrolyzed during incubation with HCl at 45˚C for 40 min, generating MDA. One molecule of MDA reacts with two molecules of the chromogenic reactive, 1-methyl-2-phenylindole, to obtain a stable chromophore. The absorbance reading was performed on a Polaris® microplate reader (Celer Biotecnologia S. A.) at 586 nm, and data expressed in nmol/g of tissue. All reagents were purchased by Sigma-Aldrich® (St Louis, MO, USA).

## Chemicals

2,2′-Azino-bis(3-ethylbenzothiazoline-6-sulfonic acid) diammonium salt—ABTS (Chemical formula $C_{18}H_{24}N_6O_6S_4$; Sigma); 2,4,6-Tris(2-pyridyl)-s-triazine—TPTZ (Chemical formula $C_{18}H_{12}N_6$; Sigma); (±)-6-Hydroxy-2,5,7,8-tetramethylchromane-2-carboxylic acid–Trolox (Chemical formula $C_{14}H_{18}O_4$; Sigma); β-Nicotinamide adenine dinucleotide 2′-phosphate reduced tetrasodium salt hydrate—NADPH (Chemical formula $C_{21}H_{26}Na_4O_{14}P_3H_2O$; Sigma); Acetone (Chemical formula $CH_3COCH_3$; VETEC); Aluminum Chloride (Chemical formula $AlCl_3$; NEON); Calcium carbonate (Chemical formula $CaCO_3$; NEON); Catechin Hydrate (Chemical formula $C_{15}H_{14} \cdot 6H_2O$; Sigma); Chloroform (Chemical formula $CHCl_3$; VETEC); DTNB 5,5′-Dithiobis (2-nitrobenzoic acid) (Chemical formula $C_{14}H_8N_2O_8S_2$; Sigma); Ethanol (Chemical formula $C_2H_5OH$; VETEC); Ether (Chemical formula $(C2H5)2O$; CLAE J.T. Baker—Plillipsburg, USA); Ferric chloride hexahydrate (Chemical formula $FeCl_3 \cdot 6H_2O$; NEON); Folin–Ciocalteu's phenol reagente (Sigma); Gallic Acid (Chemical

formula (HO)3C6H2CO2H; Sigma); Glacial Acetic acid (Chemical formula $CH_3CO_2H$; VETEC); Glutathione reductase (Chemical formula $C_{10}H_{17}N_3O_6S$; Sigma); Hexane (Chemical formula C6H14; CLAE J.T. Baker—Plillipsburg, USA); Hydrochloric Acid (Chemical formula HCL; NEON); Methanol (Chemical formula CH4O; CLAE J.T. Baker—Plillipsburg, USA); Potassium hidroxide (Chemical formula KOH; VETEC); Potassium Persulfate (Chemical formula $K_2S_2O_{8}$; NEON); Sodium acetate (Chemical formula $CH_3COONa$; NEON); Sodium carbonate (Chemical formula $Na2CO_3$; NEON); Sodium hydroxide (Chemical formula NaOH; NEON); Sodium Nitrite (Chemical formula $NaNO_2$; NEON); Sodium sulfate (Chemical formula Na2SO4; F. MAIA); Trichloroacetic acid–TCA (Chemical formula $C_2HCl_3O_2$; Sigma).

## Statistical analysis

The results were expressed as mean ± standard deviation. The ANOVA Two Way Analysis of Variance test was used, followed by the Tukey post-test. Statistically significant differences were considered with $p < 0.05$. All data were analyzed using GraphPad Prism ® version 5.01 (GraphPad Software Inc., San Diego, CA, USA).

## Results

### Determination of phenolic compounds, total antioxidant activity and fatty acid composition of macaíba palm seed kernel

The data with the determination of phenolic compounds and total antioxidant activity of macaíba palm seed kernel are reported in Table 1. The fatty acid composition of macaíba palm seed kernel is shown in Table 2.

### Effects of macaíba consumption on anxiety

**Open field test.** According to the results, DG, KG and DKG showed less ambulation than CONT (Fig 1A). The statistical analysis by two-way ANOVA showed significant effects of dyslipidemia [$F_{(1.24)} = 19.94$, $p = 0.0002$], no significant effects of kernel consumption [$F_{(1.24)} = 3.372$, $p = 0.0787$], and interaction between dyslipidemia and kernel consumption [$F_{(1.24)} = 10.67$, $p = 0.0033$].

Both groups treated with macaíba realized more rearing than CONT and DG (Fig 1B). Two-way ANOVA statistical analysis showed non-significant effects of dyslipidemia [$F_{(1.32)} = 1.716$, $p = 0.1995$], significant result by macaíba consumption [$F_{(1.32)} = 55.38$, $p < 0.0001$], and no interaction between dyslipidemia and kernel consumption [$F_{(1.32)} = 1.582$ $p = 0.2176$].

The grooming was decreased in DG and KG compared to CONT and DKG compared to all groups (Fig 1C). Two-way ANOVA statistical analysis showed significant difference of dyslipidemia [$F_{(1.52)} = 29.85$, $p < 0.0001$], kernel consumption [$F_{(1.52)} = 38.34$, $p < 0.0001$], and no interaction between dyslipidemia and kernel consumption [$F_{(1.52)} = 1.194$ $p = 0.2796$].

**Table 1. Total phenolic, flavonoids, carotenoids content and antioxidant activities of macaíba palm seed kernel.**

| Macaíba palm seed kernel | |
| --- | --- |
| Total phenolics (mg GAE/100g) | 50.90 (±0.00) |
| Total flavonoids (mg CE/100g) | 39.38 (±0.00) |
| Yellow flavonoids (mg/100g) | 0.37 (±0.01) |
| Total carotenoids (mg/100g) | 0.39 (±0.00) |
| **Total Antioxidant Activities** | |
| FRAP (µmol TE/g) | 0.06 (±0.00) |
| ABTS (µmol TE/g) | 0.85 (±0,00) |

**Table 2. Fatty acids[1] composition of macaíba palm seed kernel.**

| FATTY ACIDS | No. of Carbon Atom | MACAÍBA PALM SEED KERNEL |
|---|---|---|
| SATURATED | | |
| Caprylic | C8:0 | 4.02 |
| Pelargonic | C9:0 | 0.01 |
| Capric | C10:0 | 3.84 |
| Undecylic | C11:0 | 0.05 |
| Lauric | C12:0 | 28.50 |
| Tridecylic | C13:0 | 0.06 |
| Myristic | C14:0 | 11.28 |
| Pentadecylic | C15:0 | 0.03 |
| Palmitic | C16:0 | 9.73 |
| Margaric | C17:0 | 0.05 |
| Stearic | C18:0 | 4.93 |
| Behenic | C22:0 | 0.09 |
| Tricosylic | C23:0 | 0.02 |
| Lignoceric | C24:0 | 0.10 |
| | TOTAL | 62.71 |
| MONOUNSATURATED | | |
| Myristoleic | C14:1 ω-5 | - |
| Palmitoleic | C16:1 ω-7 | 0.06 |
| Heptadecenoic | C17:1 ω-7 | - |
| Oleic | C18:1 ω-9 | 27.74 |
| Vaccenic | C18:1 ω-7 | - |
| Gondoic | C20:1 ω-9 | 0.23 |
| Erucic | C22:1 ω-9 | - |
| | TOTAL | 28.03 |
| POLYUNSATURATED | | |
| Linoleic | C18:2 ω-6 | 2.98 |
| Alpha Linolenic | C18:3 ω-3 | - |
| Eicosadienoic | C20:2 ω-6 | - |
| Dihomo Gamma Linoleic | C20:3 ω-6 | - |
| Eicosatrienoic | C20:3 ω-3 | - |
| Arachidonic | C20:4 ω-6 | - |
| Eicosapentaenoic (EPA) | C20:5 ω-3 | - |
| Adrenic | C22:4 ω-6 | - |
| Docosapentaenoic (DPA) | C22:5 ω-3 | - |
| Docasahexaenoic (DHA) | C22:6 ω-3 | - |
| | TOTAL | 2.98 |
| UFA/SFA Ratio | | 0.49 |
| MUFA/SFA Ratio | | 0.45 |
| PUFA/SFA Ratio | | 0.05 |
| Total of fatty acid ω-9 | | 27.97 |
| Total of fatty acid ω-6 | | 2.98 |
| Total of fatty acid ω-3 | | - |

[1]Fatty acids in g/100 g of total fatty acids. SFA: Saturated fatty acids; MUFA: Monounsaturated fatty acids; PUFA: Polyunsaturated fatty acids; UFA: Unsaturated fatty acids (MUFA + PUFA).

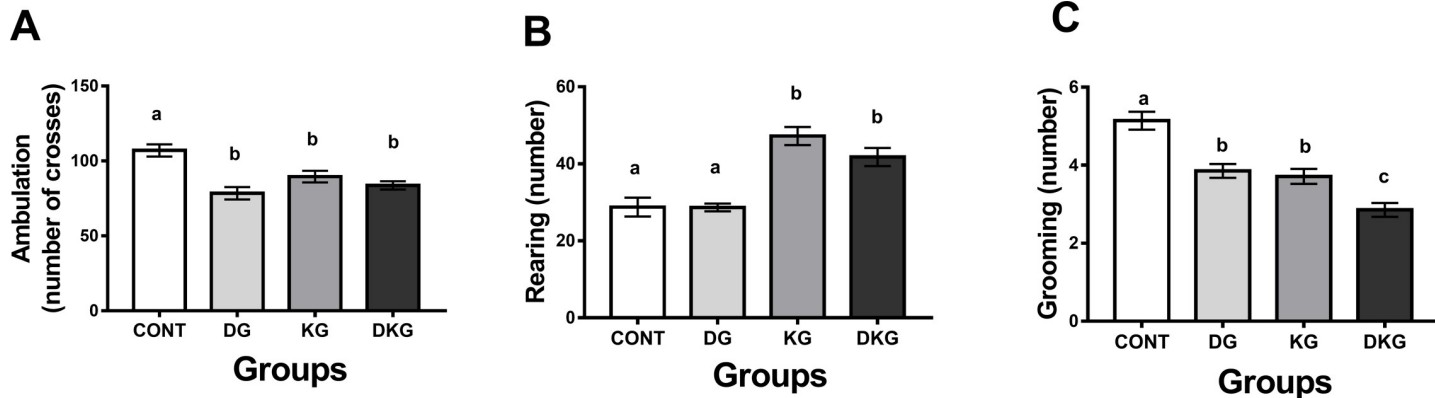

**Fig 1.** Effect of macaíba palm seed kernel on the (A) ambulation, (B) rearing, and (C) grooming parameters in the anxiety test using the open field apparatus. Values expressed as mean and standard deviation (Two way ANOVA); Legend: CONT = Control group; DG = Dyslipidemic group; KG = AFS treated group; DKG = Group treated with HFE and AFS. Different letters mean significant difference; p <0.05.

**Elevated plus maze test (EPM).** KG and DKG groups presented decreased entries in the open arms as compared to the CONT and DG groups (Fig 2A) (p <0.05). DG group also showed a reduced number of entries than CONT (p <0.05). The statistical analysis by two-way ANOVA showed significant effects of dyslipidemia [$F_{(1.44)} = 3.953$, $p = 0.05$], kernel consumption [$F_{(1.44)} = 92.62$, $p < 0.0001$], and interaction between dyslipidemia and kernel consumption [$F_{(1.44)} = 327.6$, $p = 0.008$].

Both groups treated with macaíba palm seed kernel (KG and DKG) spent more time in open arms (p <0.05) as compared to CONT and DG groups (Fig 2B). The statistical analysis of EPM data by two-way ANOVA showed significant effects of dyslipidemia [$F_{(1.52)} = 210.5$, $p < 0.0001$], kernel consumption [$F_{(1.52)} = 1257$, $p < 0.0001$], and interaction between dyslipidemia and kernel consumption [$F_{(1.52)} = 327.6$, $p < 0.0001$].

Consistent with these results, DKG group stayed longer in the central area when compared to all groups (p <0.05) (Fig 2C). Two-way ANOVA statistical analysis showed significant effects of dyslipidemia [$F_{(1.44)} = 27.92$, $p < 0.0001$], kernel consumption [$F_{(1.44)} = 14.5$, $p = 0.0004$], and no interaction between dyslipidemia and kernel consumption [$F_{(1.44)} = 0.8694$ $p = 0.3562$].

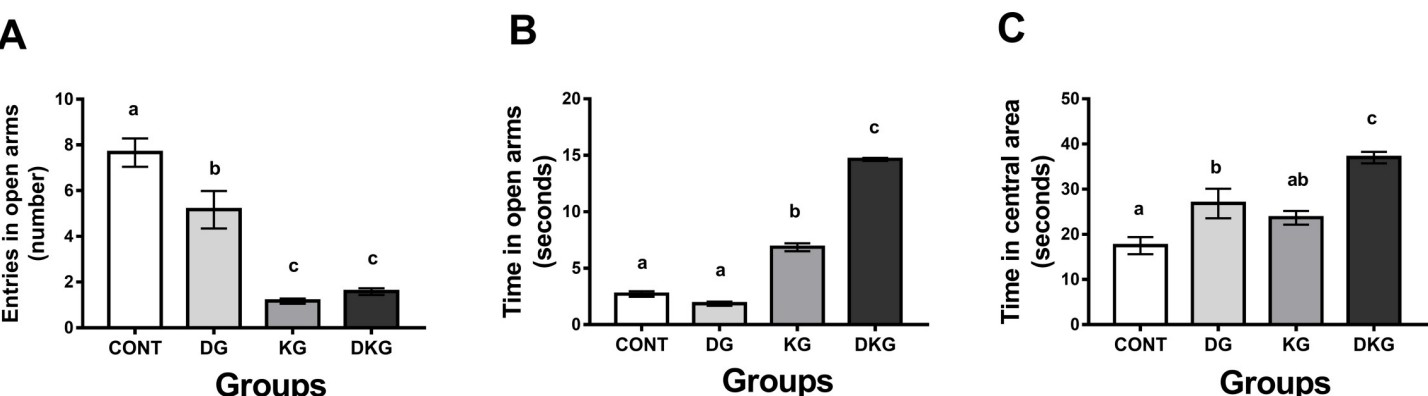

**Fig 2.** Effect of macaíba palm seed kernel on (A) number of entries in the open arms, (B) time spent in the open arms, and (C) time spent in the central area using the elevated plus maze apparatus. Values expressed as mean and standard deviation (Two way ANOVA); Legend: CONT = Control group; DG = Dyslipidemic group; KG = AFS treated group; DKG = Group treated with HFE and AFS. Different letters mean significant difference; p <0.05.

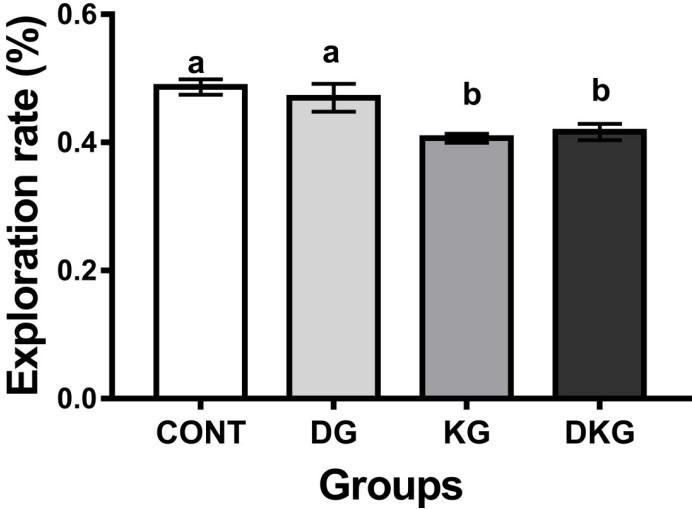

**Fig 3. Effect of macaíba palm seed kernel on the exploration rate in the short-term memory test.** Values expressed as mean and standard deviation (Two way ANOVA); Legend: CONT = Control group; DG = Dyslipidemic group; KG = AFS treated group; DKG = Group treated with HFE and AFS. Different letters mean significant difference; p <0.05.

**Object recognition test.**    As for the object recognition test, a reduction in the rate of exploration of the new object was observed in the KG and DKG groups when compared to CONT and DG (Fig 3). Two-way ANOVA statistical analysis showed non-significant effects of dyslipidemia [$F_{(1.44)}$ = 0.8502, p = 0.3615], significant effects of kernel consumption [$F_{(1.44)}$ = 21.26, p < 0.0001], and no interaction between dyslipidemia and kernel consumption [$F_{(1.44)}$ = 0.8502 p = 0.8188].

## Oxidative stress markers

**Total glutathione and malondialdehyde (MDA) content.**    The brains of dyslipidemic animals treated with macaíba—the DKG group presented a higher concentration of glutathione than the KG group (p <0.05) (Fig 4A). As for MDA levels in the animals' brains, the animals treated with macaíba palm seed kernel—the KG and DKG groups presented significant reductions as compared to the CONT and DG groups (p <0.05) (Fig 4B).

## Fatty acid composition of the brain

According to the fatty acid profile results for the animals' brains (Table 3), it can be seen that in relation to saturated fatty acids, the CONT group presented a greater amount of myristic acid in the brain as compared to the DG group, and the CONT, KG, and DKG groups presented higher values of palmitic acid compared to the DG group. Stearic acid was also found in the animals' brains, and the animals in the KG and DKG groups presented higher concentrations than the DG group. For monounsaturated fatty acids, the DG group presented higher values of elaidic and eicosanoic acids as compared to the KG and DKG groups. As for palmitoleic acid, the DG group presented higher concentrations compared to the CONT and KG groups.

Higher values of oleic acid were found in the CONT and DG groups, in relation to the KG and DKG groups. For polyunsaturates, the CONT and DG groups presented a greater amount of linoleic acid in the brain tissue, (the CONT group as compared to the KG group, and the DG group as compared to the KG and DKG groups). Eicosatetraenoic acid was also found in

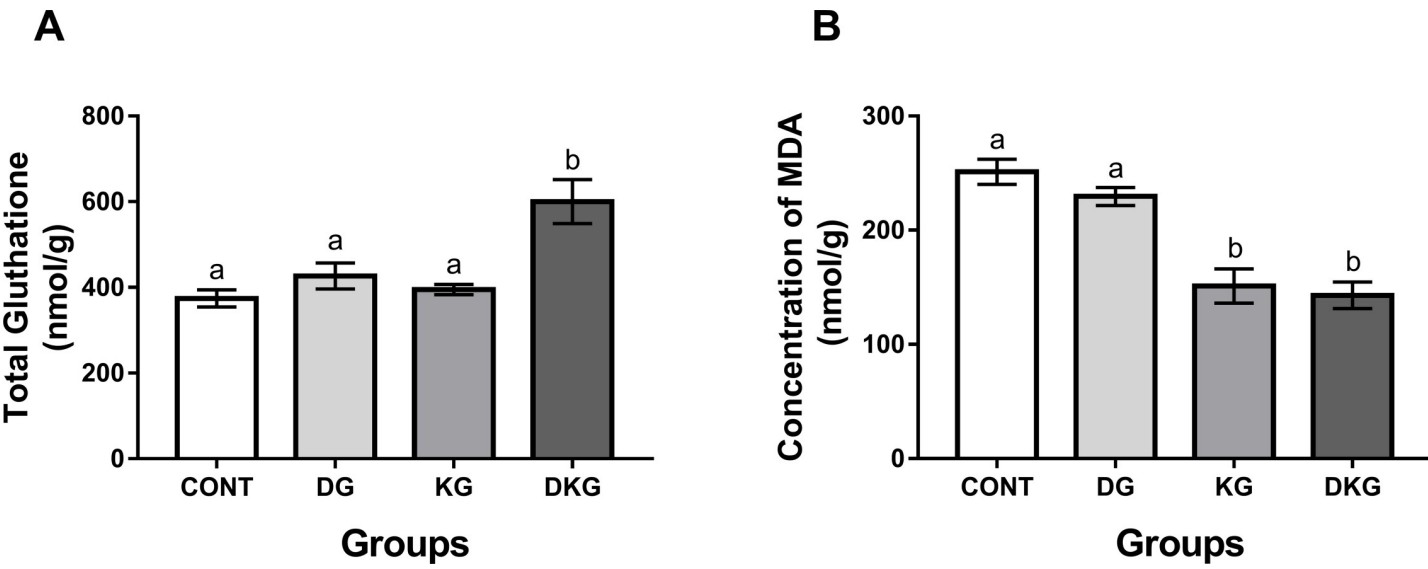

**Fig 4. MDA and GSH levels in the brain of rats treated with macaíba palm seed kernel.** Values expressed as mean and standard deviation (Two way ANOVA); Legend: CONT = Control group; DG = Dyslipidemic group; KG = AFS treated group; DKG = Group treated with HFE and AFS. Different letters mean significant difference; $p < 0.05$.

the animals' brains, and the KG and DKG groups presented higher values than the DG group. When compared to the CONT and DG groups, the groups supplemented with macaíba presented lower values of linoleic and eicosatetraenoic acids in brain tissue.

**Table 3. Brain fatty acids composition of rats treated with macaíba palm seed kernel.**

| | | | GROUPS | | |
|---|---|---|---|---|---|
| FATTY ACIDS | NO. OF CARBON | CONT | DG | KG | DKG |
| SATURATED | | | | | |
| Myristic acid | (14:0) | 1.32 ± 0.01a | 1.10±0.01b | 1.25±0.22ab | 1.28±0.03ab |
| Palmitic acid | (16:0) | 22.94±1.21a | 21.96±0.02b | 23.17±0.17a | 23.16±0.03a |
| Stearic acid | (18:0) | 21.87±0.01a | 21.69±0.01a | 22.26±0.13b | 22.27±0.21b |
| | TOTAL | 46.13 | 44.75 | 46.68 | 46.71 |
| MUFAS | | | | | |
| Palmitoleic acid | (16:1) | 1.10±0.05a | 1.28±0.01b | 1.01±0.13a | 1.15±0.07ab |
| Elaidic acid | (18:1n9t) | 5.57±0.20ab | 5.89±0.02a | 5.51±0.14b | 5.45±0.18b |
| Oleic acid | (18:1n9c) | 22.50±0.08ab | 23.61±0.02a | 21.90±0.46b | 21.39±0.69b |
| Eicosanoic acid | (20:1) | 3.88±0.23ab | 4.26±0.01a | 3.75±0.32b | 3.72±0.05b |
| | TOTAL | 33.05 | 35.04 | 32.17 | 31.71 |
| PUFAS | | | | | |
| Linoleic acid | (18:2n6c) | 1.94±0.02a | 2.14±0.02a | 1.43±0,07b | 1.39±0,14b |
| Eicosatetraenoic acid | (20:4n6) | 14.31±0.53ab | 13. ±0.01a | 14,80±0,84b | 15.39±0.70b |
| Docosahexaenoic acid | (22:6n3) | 4.56±0.32a | 4.78±0.01a | 4.92±0.19a | 4.79±0.17a |
| | TOTAL | 20.81 | 20.20 | 21.15 | 21.57 |
| AGP/AGS | | 0.45 | 0.45 | 0.45 | 0.46 |

CONT: Control group; DG: Dislipidemic group; KG: Treated with macaíba palm seed kernel; DKG: Dislipidemic rats treated with macaíba palm seed kernel. MUFAS: Monounsaturated fatty acids; PUFAS: Polyunsaturated fatty acids. One Way Anova, $p<0,05$. Different letters mean significant difference.

## Discussion

In the present study, the parameters of anxiety, memory, and lipid peroxidation in the brains of healthy and dyslipidemic rats treated with macaíba were analyzed. The results indicate that macaíba palm seed kernel, when offered to healthy and dyslipidemic rats, reduces anxiety and decreases oxidative stress in the brain of the animals, this, due to the reduction of lipid peroxidation. However, the diet promoted lower memory performance in the animals during the object recognition test.

To evaluate the anxiety parameters in the present study we used the open field and the elevated plus maze. In the open field higher rearings and less grooming in the groups treated macaíba was observed. Grooming is related to aversive situations. Anxious animals groom more often [42]. Warneke et al., [43] treated rats with a cafeteria diet and observed an increase in grooming in adult rats when they were exposed to EPM and no difference when used OF. Cafeteria diet is composed of palatable foods, such as cookies, chocolate and bread, which are not sources of antioxidants compounds such as macaíba palm seed kernel.

In the present work we also exposed the rats to the EPM. The EPM is a widely used anxiety model [44–46], and has been validated by Pellow et al. [35] in behavioral, physiological, and pharmacological approaches. The DKG group animals stayed longer in the central area as compared to all groups. The KG and DKG animals presented fewer entries into the open arms, but the time spent in the open arms was longer compared to CONT and DG. Rodents present an aversion to open and unprotected spaces, and animals that spend more time in open and unprotected spaces exhibit anxiolytic-like behavior [47]. A similar result was observed in a study that treated hypercholesterolemic animals with grape seed extract for four months. The EPM test demonstrated that grape seed extract increases the time spent in the open arms as compared to the hypercholesterolemic rats [48]. Both the grape seed and the macaíba palm seed kernel are sources of antioxidant compounds, besides that, macaíba palm seed kernel presents a variety of lipids.

An increase in oxidative stress could induce abnormalities and changes in membrane lipids and proteins [49]. Anxiety, as well as other neurobehavioral changes, can be mediated by oxidative stress damages in the brain [50]. Oxidative stress is an important factor in development of neurodegenerative and neuropsychiatric diseases, including stress and anxiety [51]. Due to the brain's oxygen consumption, relatively low antioxidant defenses, and high fat content, this tissue is very susceptible to damage caused by oxidative stress [52]. Considering the role of oxidative stress in anxiety-like behavior, oxidative stress markers and antioxidant concentrations have been evaluated in various animal studies [53]. The results demonstrate that higher levels of MDA in the neural tissue are predictive of anxiogenic-like behavior [44,54].

Reduction of oxidative stress in the brain tissue of the rats was demonstrated by the decrease of MDA levels in both treated groups compared with CONT and DKG. MDA is an oxidative damage product of lipid peroxidation [41]. The DKG presented an increase in brain GSH levels compared to all groups. The components of the palm seed kernel may have potentiated this effect in DG. GSH is an essential cellular antioxidant that plays a key role in the defense of brain cells against oxidative stress [55,56]. No significant changes were observed in the levels of MDA and GSH in the DG, suggesting that the brain is not yet under cellular stress. However, treatment with macaíba palm seed kernel improved the antioxidant effect and reduced levels of oxidative stress. This also occurred in a study in which there was no increase in the production of ROS ($H_2O_2$) of the soleus muscle fibers in rats fed during a 14-day high-fat diet [57].

This decrease in oxidative stress observed in the rats' brains treated with macaíba are justified by the presence of phenolics and carotenoids in its composition. The presence of antioxidant compounds in fruits and seeds with antioxidant activity provide protection against

oxidative stress, act synergistically, and provide better protection for cellular components [58–61]. Flavonoids, in turn, act in suppressing the release of cytokines, such as IL-1β and TNF-α, control the exposure of nitric oxide and inhibit the activation of NADPH oxidase, regulating an activity of clinical and therapeutic transcription factors related to oxidation [62–64]. A similar result was observed by Batool el al. [57], in which animals were treated with almond (*Prunus amygdalus*) for 28 days. After treatment, the animals received an injection of scopolamine, a drug that reduces the antioxidant activity of certain enzymes and induces oxidative stress [65]. The almond consumption was able to reduce oxidative stress in the animals' brains, equivalent to the results found in the present study. Domínguez et al. [66] also observed that the consumption of pecan oil (*Carya illinoinensis*) by rats fed high fat diets, reduces oxidative stress. Cavalieri et al., [67] observed a significant increase in MDA and ROS production in the brain in the control and HFD (high-fat diet) groups, with the increase in the animals' age, observed up to 18 weeks of treatment. However, for the HFD group, this increase was observed earlier, starting from the third week, and the results were greater each week also in the HFD group. A significant decrease in GSH and GSH / GSSG was observed at 12 weeks in the control group. In the HFD group, this decrease was anticipated at 3 weeks.

The present study also evaluated the influence of macaíba palm seed kernel on memory tasks. Learning processes are dependent on neurogenesis, especially in the hippocampus and cortex where newly formed neurons are recruited to perform pre-existing neural activities [68,69]. In addition, increased levels of acetylcholine in the frontal cortex and hippocampus are related to memory improvement through the cholinergic system [70,71]. Evidence suggests that eicosatetraenoic acid (ARA) and docosahexaenoic acid (DHA) increase the release of acetylcholine, improving the cholinergic system which is involved in long-term potentiation, modulation processes, and synaptic plasticity [72,73]. In this study, the administration of macaíba palm seed kernel in both healthy and dyslipidemic rats led to the impairment of memory tasks in the object recognition test (ORT). ARA was found in higher concentrations in the brains of the animals receiving kernel as compared to the DG group. However there was no significant increase in DHA levels. A study [73] revealed that the MUFAs and PUFAS present in the cashew nut increase levels of DHA in animals' brains, and are associated with memory facilitation. The glutamatergic system is also believed to be involved in cognitive function improvement related to the hippocampus caused by increased DHA [74].

It was observed that the KG group presented lower levels of MUFA (2.7% and 6.7%) compared to CONT and DG groups respectively. The DKG group revealed lower levels of MUFA (4% and 9.5%) respectively compared to the CONT and DG groups. In addition to these, the content of linoleic acid in the brain was lower in both groups treated with macaíba palm seed kernel as compared to the two control groups, and revealed lower levels of oleic acid as compared to the DG group. These findings demonstrate that healthy animals treated with macaíba palm seed kernel reveal a reduction in MUFAs in the brain, which is worsened when dyslipidemia is present. Both MUFAs and PUFAs are associated with effects in multiple neurotransmitter systems; such as the glutamatergic system, the dopaminergic system, the noradrenergic system and the serotonergic system [75]. The memory loss presented in the present study can be explained by the fact that macaíba palm seed kernel does not present large amounts of MUFA or PUFA in its composition. An experimental study treating healthy animals with macaíba, which contain high concentrations of unsaturated fatty acids, revealed increased levels of acetylcholine in the brain and improved memory [76]. In contrast, diets rich in saturated fatty acids and total fat are related to lower brain levels of brain-derived neurotrophic factor (BDNF),to neuronal plasticity, and to cognitive decline [11,77]. These findings may justify the results found in the present study, in which the fatty acids present in the macaíba palm seed kernel compromised the animals' learning in the memory test.

## Conclusion

Data showed that the consumption of macaíba palm seed kernel reduced lipid peroxidation in brains and induced anxiolytic-like behaviour in health and dyslipidemic rats.

On the other hand, macaíba palm seed kernel consumption compromised the rats' learning and memory performance. To better elucidate this mechanism of memory impairment, future analyzes measuring brain-derived neurotrophic factor and neurotransmitters can be performed.

## Supporting information

**S1 Data.**
(XLSX)

## Author Contributions

**Data curation:** Roberta Cristina de França Silva, Mikaelle Albuquerque de Souza, Jaielison Yandro Pereira da Silva, Carolina da Silva Ponciano, Daline Fernandes de Souza Araújo, Marta Maria da Conceição, Celina de Castro Querino Dias.

**Formal analysis:** Vanessa Bordin Viera, Maria Elieidy Oliveira, Juliana Kessia Barbosa Soares.

**Methodology:** Vanessa Bordin Viera, Camila Carolina de Menezes Santos Bertozzo, Gerlane Coelho Guerra, Daline Fernandes de Souza Araújo, Marta Maria da Conceição, Celina de Castro Querino Dias.

**Supervision:** Camila Carolina de Menezes Santos Bertozzo, Maria Elieidy Oliveira, Juliana Kessia Barbosa Soares.

**Visualization:** Jaielison Yandro Pereira da Silva, Vanessa Bordin Viera.

**Writing – original draft:** Juliana Kessia Barbosa Soares.

**Writing – review & editing:** Mikaelle Albuquerque de Souza, Camila Carolina de Menezes Santos Bertozzo, Gerlane Coelho Guerra, Maria Elieidy Oliveira.

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
