## [Decision Letter · Decision Letter 0]

14 Jul 2020

PONE-D-20-18348

MACAÍBA ALMOND (Acrocomia intumescens Drude) CONSUMPTION REDUCES ANXIETY AND COMPROMISE MEMORY IN RAT

PLOS ONE

Dear Dr. Soares,

Thank you for submitting your manuscript to PLOS ONE. After careful consideration, we feel that it has merit but does not fully meet PLOS ONE’s publication criteria as it currently stands. Therefore, we invite you to submit a revised version of the manuscript that addresses the points raised during the review process.

Several major methodological issues are stated in the Reviewers' reports and they should be carefully addressed, while the text would benefit if edited by a native English speaker in order to avoid vernacular expressions and poor syntax. More specific comments are further stated below in the reports of two reviewers.

We look forward to receiving your revised manuscript.

Kind regards,

Branislav T. Šiler, Ph.D.

Academic Editor

PLOS ONE

Journal Requirements:

2.Thank you for stating the following in the Acknowledgments Section of your manuscript:

'The authors thank CAPES (Coordination for the Improvement of Higher Education Personnel) for the financial support and scholarship, which was a great help throughout the research.'

'This study was financed in part by the National Council for Scientific and Technological Development (CNPq)- Brazil,  Universal 2018-2.'

4. We note you have included a table to which you do not refer in the text of your manuscript. Please ensure that you refer to Table 2 in your text; if accepted, production will need this reference to link the reader to the Table.

Reviewers' comments:

Reviewer's Responses to Questions

**Comments to the Author**

1. Is the manuscript technically sound, and do the data support the conclusions?

Reviewer #1: No

Reviewer #2: Partly

2. Has the statistical analysis been performed appropriately and rigorously? 

Reviewer #1: No

Reviewer #2: No

3. Have the authors made all data underlying the findings in their manuscript fully available?

Reviewer #1: Yes

Reviewer #2: Yes

4. Is the manuscript presented in an intelligible fashion and written in standard English?

Reviewer #1: No

Reviewer #2: No

5. Review Comments to the Author

Reviewer #1: 1) Title

The title did not reflect the nature of the study and as seen from the manuscript the effectiveness of macaíba almonds was evaluated on anxiolytic activity, memory preservation and oxidative stress in the brain of dyslipidemic rats.

Hence, the title should read:

- Evaluation of the effectiveness of macaíba almonds on anxiolytic activity, memory preservation and oxidative stress in the brain of dyslipidemic rats.

2) Abstract

The Abstract did not provide an informative and quantitative summary of the research undertaken while the Conclusion was not stated in the Abstract.

3) Introduction

The problem statement was not clearly spelt out. A clear justification on the importance of this study should be deliberated.

4) Materials and Methods

➢ All the chemicals used in the study should be named in this section.

5) Results

Results are not well deliberated.

➢ Where is the description for below section in the manuscript:

3.4 Determination of phenolic compounds and total antioxidant activity

6) Discussion

Author need to discuss about the Results obtained rather that used the general sentences. There is no possible explanation for findings and no conclusion with implication for each section of the work. Please revise the relevance of the findings in the light of other comparable studies.

7) Conclusion

The Conclusion drawn in this study is not clear nor convincing. Authors should also include in this section: constrains/limitations of the present study and future studies to be undertaken. The conclusion should provide a place to persuasively and succinctly restate the research problem.

8) Figures

➢ Figures What is X and Y axis? Each axis should be labeled with name and unit.

9) English language usage

The poorly worded/construction of the sentences should be improved through the text by a native English speaker to ensure the use of standard and correct English.

10) Please rectify the below throughout the manuscript:

- (e.g. Author stated: .....reflux for 4 min --------a further 3 minutes...Which one is correct?? min or minutes)

➢ Authors did not adhere to the format of the Journal

➢ Please check for:

[41] (BUDZYNSKA et al., 2015).

[45] (BASIRICO et al, 2017).

Reviewer #2: 1. The article is not written in a proper English language The article should be re-written with the help of English language expert.

2. How did the rat administer with almond. what is almond flour solution. How it was prepared. Add the details.

3. Describe the composition of HFE and its method of preparation

4. It is not clear whether the administration of HFE was continued through out the experimental duration or it was only given before the supplementation of almond for 14 days.

5. Mention the recorded parameter of test at the end of each test.

6. In Object recognition test, the nature of object A1, A2, and A3 is not clear, whether all three were different ot there was any similarity existed between the objects.

7. The experimental protocol involve two factors; therefore the data should re-analyzed by two-way ANOVA.

8. Since authors are explaining compromised memory in term of lipid content of almond, so the lipid content of almond should be analyzed.

9. Merge the small paragraphs to make comprehensive para.

6. PLOS authors have the option to publish the peer review history of their article (what does this mean?). If published, this will include your full peer review and any attached files.

Reviewer #1: No

Reviewer #2: No

---

## [Author Response · Author response to Decision Letter 0]

8 Oct 2020

I would like to amend my financial disclosure: “This study was supported by CNPQ (National Council for Scientific and Technological Development) Brazil - Universal 423166/2018-2. The funders had no role in study design, data collection and analysis, decision to publish, or preparation of the manuscript. JKBS is the researcher responsible for using the funding. There was no additional external funding received for this study.”

It was inserted in a cover letter.

---

## [Decision Letter · Decision Letter 1]

26 Oct 2020

PONE-D-20-18348R1

EVALUATION OF THE EFFECTIVENESS OF MACAÍBA ALMONDS (Acrocomia intumescens Drude) ON ANXIOLYTIC ACTIVITY, MEMORY PRESERVATION AND OXIDATIVE STRESS IN THE BRAIN OF DYSLIPIDEMIC RATS

PLOS ONE

Dear Dr. Soares,

Thank you for submitting your manuscript to PLOS ONE. After careful consideration, we feel that it has merit but does not fully meet PLOS ONE’s publication criteria as it currently stands. Therefore, we invite you to submit a revised version of the manuscript that addresses the points raised during the review process.

The modern science does not mention such an expression like "Macaíba almonds". As I am informed, *Acrocomia intumescens* is locally known as Macaíba palm, and seed kernel of a palm cannot be termed "almond". I suggest using "Macaíba palm seed kernel" (in the main title and thoughout the text; figures and tables too).

I also strongly advice the authors to engage a professional language editing agency, since the text still remains hardly readable due to weak syntax and poor grammar. I also suggest consulting a senior reseacher to meticolously check and rectify non-scientific expressions and vague sentences towards clarifying their meaning. The manuscript will not be publishable in this form.

We look forward to receiving your revised manuscript.

Kind regards,

Branislav T. Šiler, Ph.D.

Academic Editor

PLOS ONE

Reviewers' comments:

Reviewer's Responses to Questions

**Comments to the Author**

1. If the authors have adequately addressed your comments raised in a previous round of review and you feel that this manuscript is now acceptable for publication, you may indicate that here to bypass the “Comments to the Author” section, enter your conflict of interest statement in the “Confidential to Editor” section, and submit your "Accept" recommendation.

Reviewer #2: (No Response)

2. Is the manuscript technically sound, and do the data support the conclusions?

Reviewer #2: Yes

3. Has the statistical analysis been performed appropriately and rigorously? 

Reviewer #2: No

4. Have the authors made all data underlying the findings in their manuscript fully available?

Reviewer #2: Yes

5. Is the manuscript presented in an intelligible fashion and written in standard English?

Reviewer #2: No

6. Review Comments to the Author

Reviewer #2: Comments

1. In introduction authors are stating “There are few studies assessing the potential and quality of macaíba almonds, and include possible effects of their ingestion” it is better to mention those studies and then mention the purpose of the current study. This would clearly highlight the novelty of the study.

2. Mention the source of chemicals

3. What are ABTS and TPC? Abbreviations are still not properly defined. Define the abbreviations at the first place and use them consistently.

4. The universal unit for the speed of centrifugation is g (gravitational force). Mention all the “rpm” value in “g” values.

5. Authors have analyzed the results by two-way ANOVA but they are not mentioning the statistical effect of two factors (dyslipidemia and almond treatment). I would suggest to follow the pattern of writing all results as follow:

Elevated plus maze test (EPM)

The statistical analysis of EPM data by two-way ANOVA showed significant/non-significant effects of dyslipidemia [F(df) = ….., p = …], macaíba consumption [F(df) = ….., p = …], and interaction between dyslipidemia and macaíba consumption [F(df) = ….., p = …]. AG and DAG groups presented decreased entries in the open arms in as compared to the CONT and DG groups (Figure 2A) (p <0.05). DG group also showed reduced number of entries than CONT (p <0.05). Both groups treated with macaiba almond (AG and DAG) spent more time in open arms (p <0.05) as compared to ……???? group (Figure 2B). Consistent with these results, DAG group stayed longer in the central area when compared to all groups (p <0.05) (Figure 2C).

6. What is “Test-t” mentioned in the legends. Authors seem to mention Tukey’s test here. If it is so, eliminate “Test-t” from legends and write Tukey’s test completely.

7. The article still need English language correction.

7. PLOS authors have the option to publish the peer review history of their article (what does this mean?). If published, this will include your full peer review and any attached files.

Reviewer #2: No

---

## [Author Response · Author response to Decision Letter 1]

12 Nov 2020

Reviewer #2: Comments

1. In introduction authors are stating “There are few studies assessing the potential and quality of macaíba almonds, and include possible effects of their ingestion” it is better to mention those studies and then mention the purpose of the current study. This would clearly highlight the novelty of the study.

Reply: Thank you. It was removed. 

“There are few studies assessing the potential and quality of macaíba pulp and oil [22, 24]. A research confirmed the anti-inflammatory and diuretic activities of the pulp’s oil in rats [25]. However, there is no scientific evidence of the effect in human or animal health induced by the consumption of macaiba almonds.

Based on the above, it is expected that the macaíba almond will present protective and antioxidant effects, with anxiolytic activity and memory preservation in health rats or with dyslipidemia. The aim of this study was to evaluate the effects of macaíba almonds on anxiety behavior and short-term memory in adult health and dyslipidemic Wistar rats.”

2. Mention the source of chemicals

Reply: Thank you. It was inserted.

Chemicals

2,2′-Azino-bis(3-ethylbenzothiazoline-6-sulfonic acid) diammonium salt - ABTS (Chemical formula C18H24N6O6S4; Sigma); 2,4,6-Tris(2-pyridyl)-s-triazine - TPTZ (Chemical formula C18H12N6; Sigma); (±)-6-Hydroxy-2,5,7,8-tetramethylchromane-2-carboxylic acid – Trolox (Chemical formula C14H18O4; Sigma); β-Nicotinamide adenine dinucleotide 2′-phosphate reduced tetrasodium salt hydrate - NADPH (Chemical formula C21H26Na4O14P3H2O; Sigma); Acetone (Chemical formula CH3COCH3; VETEC ); Aluminum Chloride (Chemical formula AlCl3; NEON); Calcium carbonate (Chemical formula CaCO3; NEON); Catechin Hydrate (Chemical formula C15H14. 6H2O; Sigma); Chloroform (Chemical formula CHCl₃; VETEC); DTNB 5,5′-Dithiobis (2-nitrobenzoic acid) (Chemical formula C14H8N2O8S2; Sigma); Ethanol (Chemical formula C2H5OH; VETEC); Ether (Chemical formula (C2H5)2O; CLAE J.T. Baker - Plillipsburg, USA); Ferric chloride hexahydrate (Chemical formula FeCl3·6H2O; NEON); Folin–Ciocalteau’s phenol reagente (Sigma); Gallic Acid (Chemical formula (HO)3C6H2CO2H; Sigma); Glacial Acetic acid (Chemical formula CH3CO2H; VETEC); Glutathione reductase (Chemical formula C10H17N3O6S ; Sigma); Hexane (Chemical formula C6H14; CLAE J.T. Baker - Plillipsburg, USA); Hydrochloric Acid (Chemical formula HCL; NEON); Methanol (Chemical formula CH4O; CLAE J.T. Baker - Plillipsburg, USA); Potassium hidroxide (Chemical formula KOH; VETEC); Potassium Persulfate (Chemical formula K2S208; NEON); Sodium acetate (Chemical formula CH3COONa; NEON); Sodium carbonate (Chemical formula Na2CO3; NEON); Sodium hydroxide (Chemical formula NaOH; NEON); Sodium Nitrite (Chemical formula NaNO2; NEON); Sodium sulfate (Chemical formula Na2SO4; F. MAIA); Trichloroacetic acid –TCA (Chemical formula C2HCl3O2; Sigma). 

3. What are ABTS and TPC? Abbreviations are still not properly defined. Define the abbreviations at the first place and use them consistently.

Reply: Thank you. It was inserted.

“The ABTS (2,2'-Azino-Bis (3-Ethylbenzothiazoline-6-Sulfonic Acid) Diammonium Sal) method was performed according to the methodology described by Surveswaran et al. [30], with modifications.

The absorbance of the extract was compared with a standard curve of gallic acid to estimate the total phenolic compound (TPC)…”

4. The universal unit for the speed of centrifugation is g (gravitational force). Mention all the “rpm” value in “g” values.

Reply: Thank you. It was modified.

MDA: “Then, placed in a water bath, with agitation, at 45 ° C, for 40 minutes and, subsequently, taken to centrifugation at 2500 xg, for 5 minutes, at 4°C.”

Total Carotenoids: “Subsequently, centrifugation was performed at 5724 xg for 10 minutes for later reading using a spectrophotometer (BEL Photonics, Piracicaba, São Paulo, Brazil) at 450 nm.”

Total yellow flavonoids: “Afterwards, centrifugation was performed at 7244 xg for 15 minutes, for later reading with a spectrophotometer (BEL Photonics, Piracicaba, São Paulo, Brazil) at 374 nm.”

5. Authors have analyzed the results by two-way ANOVA but they are not mentioning the statistical effect of two factors (dyslipidemia and almond treatment). I would suggest to follow the pattern of writing all results as follow:

Elevated plus maze test (EPM)

The statistical analysis of EPM data by two-way ANOVA showed significant/non-significant effects of dyslipidemia [F(df) = ….., p = …], macaíba consumption [F(df) = ….., p = …], and interaction between dyslipidemia and macaíba consumption [F(df) = ….., p = …]. AG and DAG groups presented decreased entries in the open arms in as compared to the CONT and DG groups (Figure 2A) (p <0.05). DG group also showed reduced number of entries than CONT (p <0.05). Both groups treated with macaiba almond (AG and DAG) spent more time in open arms (p <0.05) as compared to ……???? group (Figure 2B). Consistent with these results, DAG group stayed longer in the central area when compared to all groups (p <0.05) (Figure 2C).

Thank You. It was inserted: 

“Open field test

 According to the results, DG, AG and DAG showed less ambulation than CONT (Figure 1A). The statistical analysis by two-way ANOVA showed significant effects of dyslipidemia [F(1.24) = 19.94, p =0.0002], no significant effects of macaíba consumption [F(1.24) = 3.372, p = 0.0787], and interaction between dyslipidemia and macaíba consumption [F(1.24) = 10.67, p =0.0033]. 

Both groups treated with macaiba realized more rearing than CONT and DG (Figure 1B). Two-way ANOVA statistical analysis showed non-significant effects of dyslipidemia [F(1.32) = 1.716, p =0.1995], significant result by macaíba consumption [F(1.32) = 55.38, p < 0.0001], and no interaction between dyslipidemia and macaíba consumption [F(1.32) = 1.582 p = 0.2176].

The grooming was decreased in DG and AG compared to CONT and DAG compared to all groups (Figure 1C). Two-way ANOVA statistical analysis showed significant difference of dyslipidemia [F(1.52) = 29.85, p <0.0001], macaíba consumption [F(1.52) = 38.34, p < 0.0001], and no interaction between dyslipidemia and macaíba consumption [F(1.52) = 1.194 p = 0.2796].

Elevated plus maze test (EPM)

AG and DAG groups presented decreased entries in the open arms in as compared to the CONT and DG groups (Figure 2A) (p <0.05). DG group also showed reduced number of entries than CONT (p <0.05). The statistical analysis by two-way ANOVA showed significant effects of dyslipidemia [F(1.44) = 3.953, p =0.05], macaíba consumption [F(1.44) = 92.62, p < 0.0001], and interaction between dyslipidemia and macaíba consumption [F(1.44) = 327.6, p =0.008].

Both groups treated with macaiba almond (AG and DAG) spent more time in open arms (p <0.05) as compared to CONT and DG groups (Figure 2B). The statistical analysis of EPM data by two-way ANOVA showed significant effects of dyslipidemia [F(1.52) = 210.5, p < 0.0001], macaíba consumption [F(1.52) = 1257, p < 0.0001], and interaction between dyslipidemia and macaíba consumption [F(1.52) = 327.6, p < 0.0001]. 

Consistent with these results, DAG group stayed longer in the central area when compared to all groups (p <0.05) (Figure 2C). Two-way ANOVA statistical analysis showed significant effects of dyslipidemia [F(1.44) = 27.92, p < 0.0001], macaíba consumption [F(1.44) = 14.5, p = 0.0004], and no interaction between dyslipidemia and macaíba consumption [F(1.44) = 0.8694 p =0.3562].

Object recognition test

As for the object recognition test, a reduction in the rate of exploration of the new object was observed in the AG and DAG groups treated with macaíba almonds when compared to CONT and DG (Figure 3). Two-way ANOVA statistical analysis showed non-significant effects of dyslipidemia [F(1.44) = 0.8502, p = 0.3615], significant effects of macaíba consumption [F(1.44) = 21.26, p < 0.0001], and no interaction between dyslipidemia and macaíba consumption [F(1.44) = 0.8502 p = 0.8188].”

6. What is “Test-t” mentioned in the legends. Authors seem to mention Tukey’s test here. If it is so, eliminate “Test-t” from legends and write Tukey’s test completely.

Thank you, it was removed.

---

## [Editor Report · Decision Letter 2]

17 Nov 2020

PONE-D-20-18348R2

EVALUATION OF THE EFFECTIVENESS OF MACAÍBA ALMONDS (Acrocomia intumescens Drude) ON ANXIOLYTIC ACTIVITY, MEMORY PRESERVATION AND OXIDATIVE STRESS IN THE BRAIN OF DYSLIPIDEMIC RATS

PLOS ONE

Dear Dr. Soares,

Thank you for submitting your manuscript to PLOS ONE. After careful consideration, we feel that it has merit but does not fully meet PLOS ONE’s publication criteria as it currently stands. Therefore, we invite you to submit a revised version of the manuscript that addresses the points raised during the review process.

The authors have failed to address the Academic Editor conerns raised in the previous review round. Moreover, in their Response to reviewers, they did not provide answers for Editor's concerns, so I may assume they have somehow overlooked them. The authors are advised to read the previous report carefully and rectify the manuscript accordingly.

We look forward to receiving your revised manuscript.

Kind regards,

Branislav T. Šiler, Ph.D.

Academic Editor

PLOS ONE

---

## [Author Response · Author response to Decision Letter 2]

10 Dec 2020

Editor’s comment

The modern science does not mention such an expression like "Macaíba almonds". As I am informed, Acrocomia intumescens is locally known as Macaíba palm, and seed kernel of a palm cannot be termed "almond". I suggest using "Macaíba palm seed kernel" (in the main title and thoughout the text; figures and tables too).

I also strongly advice the authors to engage a professional language editing agency, since the text still remains hardly readable due to weak syntax and poor grammar. I also suggest consulting a senior reseacher to meticolously check and rectify non-scientific expressions and vague sentences towards clarifying their meaning. The manuscript will not be publishable in this form.

Reply: Thank you. 

The title was modified: EVALUATION OF THE EFFECTIVENESS OF MACAÍBA PALM SEED KERNEL (Acrocomia intumescens Drude) ON ANXIOLYTIC ACTIVITY, MEMORY PRESERVATION AND OXIDATIVE STRESS IN THE BRAIN OF DYSLIPIDEMIC RATS

The Almond group (AG) was modified to Macaiba group (MG) and dyslipidemic almond (DAG) was modified to dyslipidemic Macaiba group (DMG)

Macaíba palm seed kernel was inserted thoughout the text, figures and tables 

The manuscript was revised by David Hardind (declaration is in the end of this response) 

Reviewers' comments:

Reviewer #2: 

Reviewer #2: Comments

1. In introduction authors are stating “There are few studies assessing the potential and quality of macaíba almonds, and include possible effects of their ingestion” it is better to mention those studies and then mention the purpose of the current study. This would clearly highlight the novelty of the study.

Reply: Thank you. It was removed. 

“There are few studies assessing the potential and quality of macaíba pulp and oil [22, 24]. A research confirmed the anti-inflammatory and diuretic activities of the pulp’s oil in rats [25]. However, there is no scientific evidence of the effect in human or animal health induced by the consumption of macaiba almonds.

Based on the above, it is expected that the macaíba almond will present protective and antioxidant effects, with anxiolytic activity and memory preservation in health rats or with dyslipidemia. The aim of this study was to evaluate the effects of macaíba almonds on anxiety behavior and short-term memory in adult health and dyslipidemic Wistar rats.”

2. Mention the source of chemicals

Reply: Thank you. It was inserted.

Chemicals

2,2′-Azino-bis(3-ethylbenzothiazoline-6-sulfonic acid) diammonium salt - ABTS (Chemical formula C18H24N6O6S4; Sigma); 2,4,6-Tris(2-pyridyl)-s-triazine - TPTZ (Chemical formula C18H12N6; Sigma); (±)-6-Hydroxy-2,5,7,8-tetramethylchromane-2-carboxylic acid – Trolox (Chemical formula C14H18O4; Sigma); β-Nicotinamide adenine dinucleotide 2′-phosphate reduced tetrasodium salt hydrate - NADPH (Chemical formula C21H26Na4O14P3H2O; Sigma); Acetone (Chemical formula CH3COCH3; VETEC ); Aluminum Chloride (Chemical formula AlCl3; NEON); Calcium carbonate (Chemical formula CaCO3; NEON); Catechin Hydrate (Chemical formula C15H14. 6H2O; Sigma); Chloroform (Chemical formula CHCl₃; VETEC); DTNB 5,5′-Dithiobis (2-nitrobenzoic acid) (Chemical formula C14H8N2O8S2; Sigma); Ethanol (Chemical formula C2H5OH; VETEC); Ether (Chemical formula (C2H5)2O; CLAE J.T. Baker - Plillipsburg, USA); Ferric chloride hexahydrate (Chemical formula FeCl3·6H2O; NEON); Folin–Ciocalteau’s phenol reagente (Sigma); Gallic Acid (Chemical formula (HO)3C6H2CO2H; Sigma); Glacial Acetic acid (Chemical formula CH3CO2H; VETEC); Glutathione reductase (Chemical formula C10H17N3O6S ; Sigma); Hexane (Chemical formula C6H14; CLAE J.T. Baker - Plillipsburg, USA); Hydrochloric Acid (Chemical formula HCL; NEON); Methanol (Chemical formula CH4O; CLAE J.T. Baker - Plillipsburg, USA); Potassium hidroxide (Chemical formula KOH; VETEC); Potassium Persulfate (Chemical formula K2S208; NEON); Sodium acetate (Chemical formula CH3COONa; NEON); Sodium carbonate (Chemical formula Na2CO3; NEON); Sodium hydroxide (Chemical formula NaOH; NEON); Sodium Nitrite (Chemical formula NaNO2; NEON); Sodium sulfate (Chemical formula Na2SO4; F. MAIA); Trichloroacetic acid –TCA (Chemical formula C2HCl3O2; Sigma). 

3. What are ABTS and TPC? Abbreviations are still not properly defined. Define the abbreviations at the first place and use them consistently.

Reply: Thank you. It was inserted.

“The ABTS (2,2'-Azino-Bis (3-Ethylbenzothiazoline-6-Sulfonic Acid) Diammonium Sal) method was performed according to the methodology described by Surveswaran et al. [30], with modifications.

The absorbance of the extract was compared with a standard curve of gallic acid to estimate the total phenolic compound (TPC)…”

4. The universal unit for the speed of centrifugation is g (gravitational force). Mention all the “rpm” value in “g” values.

Reply: Thank you. It was modified.

MDA: “Then, placed in a water bath, with agitation, at 45 ° C, for 40 minutes and, subsequently, taken to centrifugation at 2500 xg, for 5 minutes, at 4°C.”

Total Carotenoids: “Subsequently, centrifugation was performed at 5724 xg for 10 minutes for later reading using a spectrophotometer (BEL Photonics, Piracicaba, São Paulo, Brazil) at 450 nm.”

Total yellow flavonoids: “Afterwards, centrifugation was performed at 7244 xg for 15 minutes, for later reading with a spectrophotometer (BEL Photonics, Piracicaba, São Paulo, Brazil) at 374 nm.”

5. Authors have analyzed the results by two-way ANOVA but they are not mentioning the statistical effect of two factors (dyslipidemia and almond treatment). I would suggest to follow the pattern of writing all results as follow:

Elevated plus maze test (EPM)

The statistical analysis of EPM data by two-way ANOVA showed significant/non-significant effects of dyslipidemia [F(df) = ….., p = …], macaíba consumption [F(df) = ….., p = …], and interaction between dyslipidemia and macaíba consumption [F(df) = ….., p = …]. AG and DAG groups presented decreased entries in the open arms in as compared to the CONT and DG groups (Figure 2A) (p <0.05). DG group also showed reduced number of entries than CONT (p <0.05). Both groups treated with macaiba almond (AG and DAG) spent more time in open arms (p <0.05) as compared to ……???? group (Figure 2B). Consistent with these results, DAG group stayed longer in the central area when compared to all groups (p <0.05) (Figure 2C).

Thank You. It was inserted: 

“Open field test

 According to the results, DG, AG and DAG showed less ambulation than CONT (Figure 1A). The statistical analysis by two-way ANOVA showed significant effects of dyslipidemia [F(1.24) = 19.94, p =0.0002], no significant effects of macaíba consumption [F(1.24) = 3.372, p = 0.0787], and interaction between dyslipidemia and macaíba consumption [F(1.24) = 10.67, p =0.0033]. 

Both groups treated with macaiba realized more rearing than CONT and DG (Figure 1B). Two-way ANOVA statistical analysis showed non-significant effects of dyslipidemia [F(1.32) = 1.716, p =0.1995], significant result by macaíba consumption [F(1.32) = 55.38, p < 0.0001], and no interaction between dyslipidemia and macaíba consumption [F(1.32) = 1.582 p = 0.2176].

The grooming was decreased in DG and AG compared to CONT and DAG compared to all groups (Figure 1C). Two-way ANOVA statistical analysis showed significant difference of dyslipidemia [F(1.52) = 29.85, p <0.0001], macaíba consumption [F(1.52) = 38.34, p < 0.0001], and no interaction between dyslipidemia and macaíba consumption [F(1.52) = 1.194 p = 0.2796].

Elevated plus maze test (EPM)

AG and DAG groups presented decreased entries in the open arms in as compared to the CONT and DG groups (Figure 2A) (p <0.05). DG group also showed reduced number of entries than CONT (p <0.05). The statistical analysis by two-way ANOVA showed significant effects of dyslipidemia [F(1.44) = 3.953, p =0.05], macaíba consumption [F(1.44) = 92.62, p < 0.0001], and interaction between dyslipidemia and macaíba consumption [F(1.44) = 327.6, p =0.008].

Both groups treated with macaiba almond (AG and DAG) spent more time in open arms (p <0.05) as compared to CONT and DG groups (Figure 2B). The statistical analysis of EPM data by two-way ANOVA showed significant effects of dyslipidemia [F(1.52) = 210.5, p < 0.0001], macaíba consumption [F(1.52) = 1257, p < 0.0001], and interaction between dyslipidemia and macaíba consumption [F(1.52) = 327.6, p < 0.0001]. 

Consistent with these results, DAG group stayed longer in the central area when compared to all groups (p <0.05) (Figure 2C). Two-way ANOVA statistical analysis showed significant effects of dyslipidemia [F(1.44) = 27.92, p < 0.0001], macaíba consumption [F(1.44) = 14.5, p = 0.0004], and no interaction between dyslipidemia and macaíba consumption [F(1.44) = 0.8694 p =0.3562].

Object recognition test

As for the object recognition test, a reduction in the rate of exploration of the new object was observed in the AG and DAG groups treated with macaíba almonds when compared to CONT and DG (Figure 3). Two-way ANOVA statistical analysis showed non-significant effects of dyslipidemia [F(1.44) = 0.8502, p = 0.3615], significant effects of macaíba consumption [F(1.44) = 21.26, p < 0.0001], and no interaction between dyslipidemia and macaíba consumption [F(1.44) = 0.8502 p = 0.8188].”

6. What is “Test-t” mentioned in the legends. Authors seem to mention Tukey’s test here. If it is so, eliminate “Test-t” from legends and write Tukey’s test completely.

Thank you, it was removed.

---

## [Editor Report · Decision Letter 3]

15 Dec 2020

PONE-D-20-18348R3

EVALUATION OF THE EFFECTIVENESS OF MACAÍBA PALM SEED KERNEL (Acrocomia intumescens Drude) ON ANXIOLYTIC ACTIVITY, MEMORY PRESERVATION AND OXIDATIVE STRESS IN THE BRAIN OF DYSLIPIDEMIC RATS

PLOS ONE

Dear Dr. Soares,

Thank you for submitting your manuscript to PLOS ONE. After careful consideration, we feel that it has merit but does not fully meet PLOS ONE’s publication criteria as it currently stands. Therefore, we invite you to submit a revised version of the manuscript that addresses the points raised during the review process.

The authors did not adhere to the correct terminology in the text. The authors state in the second paragraph of the page 3 (I would be more precise, but the line number are not provided): "Macaíba is the fruit of the macaibeira (*Acrocomia intumescens* Drude); a palm tree that naturally occurs in northeastern Brazil." - please do not use semicolons instead of comma. Further, seed kernel extraction from the fruits is described under the subtitle "Macaiba palm seed kernel". All the experiments are performed on the seed kernel, not fruits. However, in page 4, authors describe the extraction of the constituents of "Macaíba pulp" (please do not capitalize common plant names). In this way, the reader is inductively instructed to realize that the authors have extracted compounds from fruits (above: Macaíba = fruit; then Macaíba pulp must be fruit pulp). This creates confusion of a reader. Therefore, I must insist to double-check the manuscript and to replace all the vague expressions ("Macaíba pulp", "Macaiba group") with "kernel pulp", "kernel fed group". Please do not use capitalization of random words - applicable throughout the text, subtitles and tables too; do not capitalize compound names, except for trademarks, the ones in the beginning of a sentence or in the table lines. Please uniform: is it "macaiba" or "macaíba"?

We look forward to receiving your revised manuscript.

Kind regards,

Branislav T. Šiler, Ph.D.

Academic Editor

PLOS ONE

---

## [Author Response · Author response to Decision Letter 3]

13 Jan 2021

Editor’s comment

The modern science does not mention such an expression like "Macaíba almonds". As I am informed, Acrocomia intumescens is locally known as Macaíba palm, and seed kernel of a palm cannot be termed "almond". I suggest using "Macaíba palm seed kernel" (in the main title and thoughout the text; figures and tables too).

I also strongly advice the authors to engage a professional language editing agency, since the text still remains hardly readable due to weak syntax and poor grammar. I also suggest consulting a senior reseacher to meticolously check and rectify non-scientific expressions and vague sentences towards clarifying their meaning. The manuscript will not be publishable in this form.

Reply: Thank you. 

The title was modified: EVALUATION OF THE EFFECTIVENESS OF MACAÍBA PALM SEED KERNEL (Acrocomia intumescens Drude) ON ANXIOLYTIC ACTIVITY, MEMORY PRESERVATION AND OXIDATIVE STRESS IN THE BRAIN OF DYSLIPIDEMIC RATS

The Almond group (AG) was modified to kernel group (KG) and dyslipidemic almond (DAG) was modified to dyslipidemic kernel group (DKG)

Macaíba palm seed kernel was inserted thoughout the text, figures and tables 

The manuscript was revised by David Hardind (declaration is in the end of this response) 

The authors did not adhere to the correct terminology in the text. The authors state in the second paragraph of the page 3 (I would be more precise, but the line number are not provided): "Macaíba is the fruit of the macaibeira (Acrocomia intumescens Drude); a palm tree that naturally occurs in northeastern Brazil." - please do not use semicolons instead of comma. Further, seed kernel extraction from the fruits is described under the subtitle "Macaiba palm seed kernel". All the experiments are performed on the seed kernel, not fruits. However, in page 4, authors describe the extraction of the constituents of "Macaíba pulp" (please do not capitalize common plant names). In this way, the reader is inductively instructed to realize that the authors have extracted compounds from fruits (above: Macaíba = fruit; then Macaíba pulp must be fruit pulp). This creates confusion of a reader. Therefore, I must insist to double-check the manuscript and to replace all the vague expressions ("Macaíba pulp", "Macaiba group") with "kernel pulp", "kernel fed group". Please do not use capitalization of random words - applicable throughout the text, subtitles and tables too; do not capitalize compound names, except for trademarks, the ones in the beginning of a sentence or in the table lines. Please uniform: is it "macaiba" or "macaíba"?

Thank you, it was modified. The correct is macaíba. 

Reviewers' comments:

Reviewer #2: 

Reviewer #2: Comments

1. In introduction authors are stating “There are few studies assessing the potential and quality of macaíba almonds, and include possible effects of their ingestion” it is better to mention those studies and then mention the purpose of the current study. This would clearly highlight the novelty of the study.

Reply: Thank you. It was removed. 

“There are few studies assessing the potential and quality of macaíba pulp and oil [22, 24]. A research confirmed the anti-inflammatory and diuretic activities of the pulp’s oil in rats [25]. However, there is no scientific evidence of the effect in human or animal health induced by the consumption of macaiba almonds.

Based on the above, it is expected that the macaíba almond will present protective and antioxidant effects, with anxiolytic activity and memory preservation in health rats or with dyslipidemia. The aim of this study was to evaluate the effects of macaíba almonds on anxiety behavior and short-term memory in adult health and dyslipidemic Wistar rats.”

2. Mention the source of chemicals

Reply: Thank you. It was inserted.

Chemicals

2,2′-Azino-bis(3-ethylbenzothiazoline-6-sulfonic acid) diammonium salt - ABTS (Chemical formula C18H24N6O6S4; Sigma); 2,4,6-Tris(2-pyridyl)-s-triazine - TPTZ (Chemical formula C18H12N6; Sigma); (±)-6-Hydroxy-2,5,7,8-tetramethylchromane-2-carboxylic acid – Trolox (Chemical formula C14H18O4; Sigma); β-Nicotinamide adenine dinucleotide 2′-phosphate reduced tetrasodium salt hydrate - NADPH (Chemical formula C21H26Na4O14P3H2O; Sigma); Acetone (Chemical formula CH3COCH3; VETEC ); Aluminum Chloride (Chemical formula AlCl3; NEON); Calcium carbonate (Chemical formula CaCO3; NEON); Catechin Hydrate (Chemical formula C15H14. 6H2O; Sigma); Chloroform (Chemical formula CHCl₃; VETEC); DTNB 5,5′-Dithiobis (2-nitrobenzoic acid) (Chemical formula C14H8N2O8S2; Sigma); Ethanol (Chemical formula C2H5OH; VETEC); Ether (Chemical formula (C2H5)2O; CLAE J.T. Baker - Plillipsburg, USA); Ferric chloride hexahydrate (Chemical formula FeCl3·6H2O; NEON); Folin–Ciocalteau’s phenol reagente (Sigma); Gallic Acid (Chemical formula (HO)3C6H2CO2H; Sigma); Glacial Acetic acid (Chemical formula CH3CO2H; VETEC); Glutathione reductase (Chemical formula C10H17N3O6S ; Sigma); Hexane (Chemical formula C6H14; CLAE J.T. Baker - Plillipsburg, USA); Hydrochloric Acid (Chemical formula HCL; NEON); Methanol (Chemical formula CH4O; CLAE J.T. Baker - Plillipsburg, USA); Potassium hidroxide (Chemical formula KOH; VETEC); Potassium Persulfate (Chemical formula K2S208; NEON); Sodium acetate (Chemical formula CH3COONa; NEON); Sodium carbonate (Chemical formula Na2CO3; NEON); Sodium hydroxide (Chemical formula NaOH; NEON); Sodium Nitrite (Chemical formula NaNO2; NEON); Sodium sulfate (Chemical formula Na2SO4; F. MAIA); Trichloroacetic acid –TCA (Chemical formula C2HCl3O2; Sigma). 

3. What are ABTS and TPC? Abbreviations are still not properly defined. Define the abbreviations at the first place and use them consistently.

Reply: Thank you. It was inserted.

“The ABTS (2,2'-Azino-Bis (3-Ethylbenzothiazoline-6-Sulfonic Acid) Diammonium Sal) method was performed according to the methodology described by Surveswaran et al. [30], with modifications.

The absorbance of the extract was compared with a standard curve of gallic acid to estimate the total phenolic compound (TPC)…”

4. The universal unit for the speed of centrifugation is g (gravitational force). Mention all the “rpm” value in “g” values.

Reply: Thank you. It was modified.

MDA: “Then, placed in a water bath, with agitation, at 45 ° C, for 40 minutes and, subsequently, taken to centrifugation at 2500 xg, for 5 minutes, at 4°C.”

Total Carotenoids: “Subsequently, centrifugation ws performed at 5724 xg for 10 minutes for later reading using a spectrophotometer (BEL Photonics, Piracicaba, São Paulo, Brazil) at 450 nm.”

Total yellow flavonoids: “Afterwards, centrifugation was performed at 7244 xg for 15 minutes, for later reading with a spectrophotometer (BEL Photonics, Piracicaba, São Paulo, Brazil) at 374 nm.”

5. Authors have analyzed the results by two-way ANOVA but they are not mentioning the statistical effect of two factors (dyslipidemia and almond treatment). I would suggest to follow the pattern of writing all results as follow:

Elevated plus maze test (EPM)

The statistical analysis of EPM data by two-way ANOVA showed significant/non-significant effects of dyslipidemia [F(df) = ….., p = …], macaíba consumption [F(df) = ….., p = …], and interaction between dyslipidemia and macaíba consumption [F(df) = ….., p = …]. AG and DAG groups presented decreased entries in the open arms in as compared to the CONT and DG groups (Figure 2A) (p <0.05). DG group also showed reduced number of entries than CONT (p <0.05). Both groups treated with macaiba almond (AG and DAG) spent more time in open arms (p <0.05) as compared to ……???? group (Figure 2B). Consistent with these results, DAG group stayed longer in the central area when compared to all groups (p <0.05) (Figure 2C).

Thank You. It was inserted: 

Open field test

 According to the results, DG, KG and DKG showed less ambulation than CONT (Figure 1A). The statistical analysis by two-way ANOVA showed significant effects of dyslipidemia [F(1.24) = 19.94, p =0.0002], no significant effects of kernel consumption [F(1.24) = 3.372, p = 0.0787], and interaction between dyslipidemia and kernel consumption [F(1.24) = 10.67, p =0.0033].

Both groups treated with macaíba realized more rearing than CONT and DG (Figure 1B). Two-way ANOVA statistical analysis showed non-significant effects of dyslipidemia [F(1.32) = 1.716, p =0.1995], significant result by macaíba consumption [F(1.32) = 55.38, p < 0.0001], and no interaction between dyslipidemia and kernel consumption [F(1.32) = 1.582 p = 0.2176].

The grooming was decreased in DG and KG compared to CONT and DKG compared to all groups (Figure 1C). Two-way ANOVA statistical analysis showed significant difference of dyslipidemia [F(1.52) = 29.85, p <0.0001], kernel consumption [F(1.52) = 38.34, p < 0.0001], and no interaction between dyslipidemia and kernel consumption [F(1.52) = 1.194 p = 0.2796].

PLEASE INSERT FIGURE 1 ABOUT HERE

Elevated plus maze test (EPM)

KG and DKG groups presented decreased entries in the open arms as compared to the CONT and DG groups (Figure 2A) (p <0.05). DG group also showed a reduced number of entries than CONT (p <0.05). The statistical analysis by two-way ANOVA showed significant effects of dyslipidemia [F(1.44) = 3.953, p =0.05], kernel consumption [F(1.44) = 92.62, p < 0.0001], and interaction between dyslipidemia and kernel consumption [F(1.44) = 327.6, p =0.008].

Both groups treated with macaíba palm seed kernel (KG and DKG) spent more time in open arms (p <0.05) as compared to CONT and DG groups (Figure 2B). The statistical analysis of EPM data by two-way ANOVA showed significant effects of dyslipidemia [F(1.52) = 210.5, p < 0.0001], kernel consumption [F(1.52) = 1257, p < 0.0001], and interaction between dyslipidemia and kernel consumption [F(1.52) = 327.6, p < 0.0001].

Consistent with these results, DKG group stayed longer in the central area when compared to all groups (p <0.05) (Figure 2C). Two-way ANOVA statistical analysis showed significant effects of dyslipidemia [F(1.44) = 27.92, p < 0.0001], kernel consumption [F(1.44) = 14.5, p = 0.0004], and no interaction between dyslipidemia and kernel consumption [F(1.44) = 0.8694 p =0.3562].

PLEASE INSERT FIGURE 2 ABOUT HERE

Object recognition test

As for the object recognition test, a reduction in the rate of exploration of the new object was observed in the KG and DKG groups when compared to CONT and DG (Figure 3). Two-way ANOVA statistical analysis showed non-significant effects of dyslipidemia [F(1.44) = 0.8502, p = 0.3615], significant effects of kernel consumption [F(1.44) = 21.26, p < 0.0001], and no interaction between dyslipidemia and kernel consumption [F(1.44) = 0.8502 p = 0.8188].

6. What is “Test-t” mentioned in the legends. Authors seem to mention Tukey’s test here. If it is so, eliminate “Test-t” from legends and write Tukey’s test completely.

Thank you, it was removed.

---

## [Editor Report · Decision Letter 4]

15 Jan 2021

EVALUATION OF THE EFFECTIVENESS OF MACAÍBA PALM SEED KERNEL (Acrocomia intumescens Drude) ON ANXIOLYTIC ACTIVITY, MEMORY PRESERVATION AND OXIDATIVE STRESS IN THE BRAIN OF DYSLIPIDEMIC RATS

PONE-D-20-18348R4

Dear Dr. Soares,

We’re pleased to inform you that your manuscript has been judged scientifically suitable for publication and will be formally accepted for publication once it meets all outstanding technical requirements.

Kind regards,

Branislav T. Šiler, Ph.D.

Academic Editor

PLOS ONE
---

## [Editor Report · Acceptance letter]

27 Jan 2021

PONE-D-20-18348R4 

EVALUATION OF THE EFFECTIVENESS OF MACAÍBA PALM SEED KERNEL (*Acrocomia intumescens* Drude) ON ANXIOLYTIC ACTIVITY, MEMORY PRESERVATION AND OXIDATIVE STRESS IN THE BRAIN OF DYSLIPIDEMIC RATS 

Dear Dr. Soares:

I'm pleased to inform you that your manuscript has been deemed suitable for publication in PLOS ONE. Congratulations! Your manuscript is now with our production department. 

Kind regards, 

on behalf of

Dr. Branislav T. Šiler 

Academic Editor

PLOS ONE